# Evaluating lncRNA Expression Patterns during HIV-1 Treatment Interruption

**DOI:** 10.3390/ijms24021031

**Published:** 2023-01-05

**Authors:** Tinus Schynkel, Willem van Snippenberg, Clarissa Van Hecke, Linos Vandekerckhove, Wim Trypsteen

**Affiliations:** HIV Cure Research Center, Department of Internal Medicine and Pediatrics, Ghent University and Ghent University Hospital, 9000 Ghent, Belgium

**Keywords:** lncRNA, HIV-1, treatment interruption, biomarker, HEAL, MALAT1, NEAT1, GAS5, NRON

## Abstract

Lately, the interest in long non-coding RNAs (lncRNAs) as potential drug targets and predictive markers in the context of HIV-1 has peaked, but their in vivo expression and regulation remains largely unexplored. Therefore, the present study examined lncRNA expression patterns during a clinical antiretroviral treatment interruption (ATI) trial. Peripheral blood mononuclear cells were isolated from ten patients at four timepoints: prior to ATI, 7–15 days after stop, at viral rebound and 3 months post antiretroviral therapy re-initiation. RNA was extracted and RT-qPCR on five known HIV-1-related lncRNAs (HEAL, MALAT1, NEAT1, GAS5 and NRON) was performed and correlated with HIV-1 and host marker expression. All lncRNAs correlated stronger with interferon stimulated genes (ISGs) than with HIV-1 reservoir and replication markers. However, one lncRNA, HEAL, showed significant upregulation at viral rebound during ATI compared to baseline and re-initiation of therapy (*p* = 0.0010 and *p* = 0.0094, respectively), following a similar viral-load-driven expression pattern to ISGs. In vitro knockdown of HEAL caused a significant reduction in HIV-1 infection levels, validating HEAL’s importance for HIV-1 replication. We conclude that the HIV-1-promoting lncRNA HEAL is upregulated at viral rebound during ATI, most likely induced by viral cues.

## 1. Introduction

The human immunodeficiency virus 1 (HIV-1) still caused an estimated 650,000 AIDS-related deaths in 2021 [1]. While tremendous scientific progress has been made with the development of combination antiretroviral treatment (cART), HIV-1 remains latent in viral reservoirs that are unaffected by cART and results in viral rebound upon treatment cessation. Therefore, reaching an HIV-1 cure will be crucial to end the pandemic.

Most cure strategies focus on a reduction in the reservoir, while boosting the immune system to control the remains of the virus: a shock-and-kill approach. To assess the effectiveness of a cure strategy, an ART treatment interruption (ATI) is the current gold standard to evaluate a potential delay in viral rebound or reductions in viral reservoir sizes. Although urgently needed, no strong markers have been discovered to predict viral rebound, necessitating continuous monitoring of the viral load (VL) during ATI. Increased viral loads are, however, accompanied by increased levels of inflammation [2] and an increased rate of transmission [3].

Long non-coding RNAs (lncRNAs) are a relatively new class of biomolecules that are implicated in a wide spectrum of biological processes, ranging from basal cellular functions to complex human diseases [4,5,6]. Although these RNA molecules (arbitrarily longer than 200 nucleotides) do not code for proteins, they play important regulatory roles within the cell by binding several classes of cellular factors and driving gene expression. Their transcript abundance is lower than their protein counterparts, but transcription is much more cell- and biological-process-specific, resulting in their being ideal drug target or biomarker candidates [7,8]. In the context of HIV-1 infection, several lncRNAs have already been described to play a role during HIV-1 infection in vitro, either by being hijacked by HIV-1 to aid in replication or latency, or as part of the immunological response against the virus [9]. However, lncRNA expression and their impact on HIV-1 regulation in vivo remains largely unexplored.

Nuclear-enriched abundant target 1 (NEAT1) helps to coordinate HIV-1 RNA availability during the HIV-1 replication cycle by the temporary storage of unspliced HIV-1 RNA in paraspeckle nuclear bodies [10]. The non-coding repressor of NFAT (NRON) controls HIV-1 replication through the regulation of NFAT activity during active infection [11], while targeting Tat for proteasomal degradation during latent infection [12]. HIV-1 enhanced lncRNA (HEAL) acts as a scaffold for FUS, p300 (a transcriptional coactivator) and either the HIV-1 promoter or the human Cyclin-Dependent Kinase 2 (CDK2) promotor, both contributing to HIV-1 transcription [13]. Metastasis-associated lung adenocarcinoma transcript 1 (MALAT1) was described to enhance viral transcription by retaining EZH2 from PRC2, which blocks the repressive methylation of the HIV-1 LTR [14]. Growth-arrest-specific transcript 5 (GAS5) is downregulated upon HIV-1 infection, which favors replication as GAS5 acts as a competing RNA, inhibiting the role of miR-873 [15].

In this study, we explored the in vivo expression patterns of these HIV-1-associated lncRNAs by performing quantitative PCR (qPCR) on RNA isolated from peripheral blood mononuclear cells (PBMCs) at four timepoints in a treatment interruption clinical study: before ATI under cART, during ATI (undetectable viral load), at viral rebound and 3 months after the reinitiation of cART. Next, the lncRNA expression profiles were correlated with HIV-1 viral markers and host immune response genes (i.e., restriction factors) to elucidate associations with host-driven immunity or viral reservoir dynamics in vivo, as we hypothesize that lncRNA expression might be linked to interferon-stimulated pathways and related interferon-stimulated genes (ISGs).

## 2. Results

### 2.1. Study Participants and Patient Characteristics

Ten people living with HIV-1 (PLHIV) were included and sampled at four timepoints, before and after treatment interruption (Figure 1). The included participants were adherent to cART therapy for >3 years and did not have any other medical condition upon time of inclusion. Therapy regiments followed by the participants prior to ATI were Truvada/Tivicay (4/10), Triumeq (3/10), Emtriva/Tivicay, Genvoya and Kivexa/Tivicay. CD4 nadir values were >300 cells/mm^3^ (Table 1). At least 3 months before inclusion, the participants were on a standard cART regimen including an integrase inhibitor. Included participants’ age range was between 36 and 54 with a median age of 39.5. The median time to viral rebound (TTVR) was 21 days (range: 15–36), with a median HIV-1 RNA level of log 3.30 copies/mL (range: 3.04–4.56) at viral rebound. Patient demographics can be found in Table 1. 

### 2.2. lncRNA Expression Profiles during Analytical Treatment Interruption

LncRNAs, with their unique characteristics, have the potential to play crucial roles during HIV-1 infection and could potentially be used as biomarkers. Some lncRNAs have been described in the context of HIV-1, but little is known about their expression in vivo. Therefore, the expression pattern of five known HIV-related lncRNAs (HEAL, MALAT1, NEAT1, GAS5, NRON) was determined at all four timepoints of the ATI trial.

Significant upregulation of HEAL was observed at viral rebound (T3) compared to samples prior to ATI (T1; *p* = 0.0010) and at T2 (*p* = 0.0391) (Figure 2A). Three months after cART initiation (T4), the expression of HEAL returned to baseline (*p* = 0.0094). The expression of HEAL thus follows viral replication levels. For MALAT1, significant upregulation was observed between T2 and T4, but not between T2 and T1 (*p* = 0.0312 and *p* = 0.4127, respectively), suggesting that viral rebound lowered baseline MALAT1 expression (Figure 2A). For both transcripts of NEAT1, a slight upwards trend can be observed at T2, however, no significant differences were found between any timepoints. Both GAS5 and NRON were previously shown to be downregulated during HIV-1 infection, which led to increased viral replication [15]. Here, GAS5 overall expression levels remained stable. Finally, the expression of NRON followed a pattern of limited decrease upon ATI (T2) and elevated expression at viral rebound, but none of these trends were significant (Figure 2B). Interestingly, an investigation of the expression in a previously described cohort [16,17] did not show a significant upregulation of lncRNAs in ART-naïve long-term non-progressors (LTNPs) in comparison to uninfected individuals (Appendix A).

### 2.3. Correlation between lncRNA Expression, HIV-1 Markers and TTVR Is Limited

Using previously published data from the same participants, we investigated whether the expression of the lncRNAs is correlated with markers indicative of HIV-1 reservoir and viral replication [18]. There was a limited correlation between lncRNA expression and HIV-1 markers (Figure 3A). However, three of the five lncRNAs correlated with at least one HIV-1 marker, and one transcript of NEAT1 showed significant correlations with three HIV-1 markers: Log total HIV-1 DNA copies, long LTR and Pol copies/μg RNA (Spearman r’s: 0.3189, 0.3412 and 0.3472; *p*-values: 0.0453, 0.0312 and 0.0281, respectively). HEAL was found to be correlated with Pol copies/μg RNA (Spearman r = 0.4545, *p* = 0.0032), a marker that also correlates with the viral load at T3 (Appendix A), again showing that HEAL expression follows viral replication.

Next, we evaluated the potential predictive value of expression changes in lncRNAs for the time to viral rebound. For this, Spearman correlations were calculated between the fold-change lncRNA expressions at T2 compared to baseline (T1) and the time from T2 to viral rebound (Figure 3B). Here, a borderline significant negative correlation (*p* = 0.0360) was observed between the fold change decrease in NRON and the TTVR (Figure 3B), suggesting that the drop in NRON expression might have some value in predicting the time to viral rebound. No significant correlations were found by looking at the absolute expression of the lncRNAs at T2 and the TTVR (Appendix A).

### 2.4. lncRNA Expression Shows Multiple Correlations with ISGs and Restriction Factors

We assessed potential correlations between the expression of several ISGs in these patient samples (data previously generated by De Scheerder et al. [19]) and the expression of the lncRNAs. Here, the expression of all lncRNAs correlated with the expression of ISGs (Figure 4A). HEAL correlated especially strongly with the expression pattern of ISGs like IFIT1, MX1, MX2, TRIM5 and BST2 (*p* = 0.0063, *p* = 0.0029, 0.0038, 0.0002 and 0.0119, respectively) (Figure 4B). Likewise, a repeated measurement correlation analysis performed between HEAL and these ISGs confirmed a significant correlation over the four timepoints (Appendix A) (IFIT1 *p* < 0.0001, MX1 *p* = 0.0006, MX2 *p* = 0.003, TRIM5 *p* = 0.013 and BTS2 *p* = 0.015). This indicates that HEAL expression coincides with the viral-load-driven expression patterns of interferon-stimulated genes (ISGs). This could mean that HEAL is either part of the ISG family, potentially including an immunological role, or that HEAL has a direct impact on viral replication induced by HIV-1 signaling.

### 2.5. In Vitro Knockdown of HEAL Reduces HIV-1 Infection

To assess the potential direct role of HEAL during HIV-1 infection and replication, we knocked down HEAL expression through an antisense oligo that was delivered via gymnosis to SupT1 cells. Expression levels at the moment of infection and 48 h post infection were reduced by 32.6% and 58.1%, respectively, compared to cells treated with a scrambled ASO control (Figure 5A). This knockdown induced a significant reduction in HIV-1 infection of 21.9% (*p* = 0.0006) 24 h post infection and 22.0% (*p* = 0.0002) 48 h post infection (Figure 5B), confirming previous reports that HEAL directly promotes HIV-1 replication [13]. As we used full-length NL4.3-GFP virus, infection levels at 48 h also reflect virus spread in the culture due to a second viral replication round.

## 3. Discussion

In the last two decades, with the introduction of large sequencing projects and a multitude of functional studies, lncRNAs have been recognized as key players in regulatory pathways and diseases, including HIV-1 infection [4,5,6,20]. Although ample in vitro research has led to substantial progress in understanding the role of lncRNAs during HIV-1 infection, little is known about their expression in vivo. This is a pity, as lncRNAs have a generally limited and disease-state-specific expression, and could thus be potential drug targets or biomarkers for viral rebound in treatment interruption trials.

To our knowledge, we were the first to explore the expression patterns of five well-known HIV-1-related lncRNAs in an ATI study, showing a significant upregulation of HEAL at the time of viral rebound, while its expression decreased to baseline when the HIV-1 VL became undetectable upon ART reinitiation. HEAL had previously been identified as upregulated by HIV-1 infection in monocyte-derived macrophages, microglia and T lymphocytes by Chao et al. [13]. Our findings show that HEAL expression indeed follows HIV-1 replication levels in vivo. The extent to which HEAL could be used as a predictive marker for viral rebound is unclear. Its expression did increase at the time of viral rebound in all individuals, but was not always very distinct (e.g., patients 3 and 4). Additionally, a predictive marker should be indicative ahead of time, as the goal of a predictive marker is to predict and eliminate the need for viral rebound. As HEAL was elevated at the time of viral rebound, but not significant at T2 (7–15 days post ATI) and HEAL expression at T2 was not correlated with time from T2 to viral rebound (Figure 3B), its added value as a predictive marker for measuring HIV-1 RNA will be limited.

NRON expression remained relatively stable across the ATI study, with no significant expression changes. However, the slight decrease at timepoint T2 during ATI compared to baseline expression was significantly correlated with the time from T2 to viral rebound. Follow-up research is necessary to confirm if this decline in NRON levels could have some predictive value for time to viral rebound. It will likely have to be combined with other markers to form a reliable estimate.

MALAT1 expression showed a slight but significant decrease in expression three months after treatment reinitiation at viral rebound compared to the baseline level before ATI. MALAT1 has been described by Qu et al. [14] to enhance HIV-1 transcription by retaining EZH2 from the polycomb repressive complex 2 (PRC2), which normally blocks repressive methylation of the HIV-1 LTR. Our results suggest that the event of viral rebound lowers the baseline expression of MALAT1. One could hypothesize that the lower levels of MALAT1 could make the cell population less receptive to HIV-1 replication, but this is speculative.

For the lncRNAs NEAT1, GAS5 and NRON, we were the first to assess expression in an ATI study. Jin et al. [21] reported increased MALAT1 and NEAT1 levels in ART-naïve patients compared to ART-treated patients and in vitro studies on cell lines also show that NEAT1 is upregulated following HIV-1 infection [10]. From our results, we can conclude that NEAT1 expression levels are stable in ART-treated patients, even when a short viral rebound occurs. GAS5 and NRON are known to be downregulated upon HIV-1 infection in cell line experiments [12]. However, we did not find a change in vivo expression of GAS5 or NRON when HIV-1 rebounded.

The ongoing replication or reactivation of HIV-1 affects the transcriptomic landscape of both infected and neighboring cells [22]. In our study, we assessed possible drivers of lncRNA expression changes by correlating the expression of the lncRNAs with HIV-1 reservoir markers (DNA), HIV-1 replication markers (RNA) and the expression levels of ISGs. Interestingly, correlations with HIV-1 markers were shown to be limited for the majority of lncRNAs. We observed that NEAT1 was correlated with multiple HIV-1 markers and HEAL. HEAL being the only lncRNA that was significantly upregulated at viral rebound, and that was correlated with Pol RNA, a marker that also corresponds with the viral load at T3. This might indicate that NEAT1 and HEAL expression is at least partially driven by direct viral signals, while the expression of MALAT1, GAS5 and NRON is regulated independently.

However, the expression of the lncRNAs appeared to be highly correlated with the expression of ISGs, indicating that it might be more linked to the host cellular response than to the presence of HIV-1 itself. Several lncRNAs have been reported to play a role in cellular immune responses, including NEAT1, which is involved in a positive feedback loop with RIG-I signaling and antiviral genes [23,24]. The correlation analysis of HEAL was notable, which correlated with the expression of all ISGs. Moreover, using repeated measurement analysis, we demonstrated that the expression follows the expression patterns of five ISGs (MX1, IFIT1, MX2, TRIM5 and BST2), which were previously described to be upregulated at viral rebound (T3) [18]. Overall, based on these data, it is hard to pinpoint the dominant driver of the expression of a lncRNA such as HEAL: does it have a similar expression pattern to interferon stimulated genes because HEAL itself is part of an interferon-induced response, or are both HEAL and ISGs expression independently driven by the viral load?

Chao et al. already proved that HEAL enhances HIV-1 replication [13]. In several HEAL knockdown experiments, they repeatedly showed reduced HIV-1 infection in MT4 cells. Our in vitro experiments support these findings: SupT1 cells with up to 58% HEAL knockdown were 22% less infected, showing that HEAL expression is needed for optimal HIV-1 infection. Altogether, the data suggest that HEAL is induced upon viral rebound by viral cues to aid infectivity, rather than to exert an immunological effect.

The current study has its limitations. First, this study included only 10 participants that were all male and infected with HIV-1 subtype B. Therefore, making conclusions about the wide population affected with HIV-1 is difficult. Second, the quantification of lncRNAs, ISGs, HIV-1 DNA and HIV-1 RNA was performed in PBMCs instead of the CD4 T-cell subpopulation, as in similar studies [25]. The inclusion of CD4 T-cells instead of bulk PBMCs might lead to increased levels of expression and improved detection of HIV-1 markers. Moreover, utilizing bulk PBMCs instead of sorted cell populations makes it difficult to pinpoint which cell types are responsible for lncRNA expression changes, CD4+ cells or other immune cells from the PBMC population. Finally, in recent years, two additional lncRNAs (linc01426 and lincRNA-p21) were identified to influence HIV-1 replication [26,27]. It would be of interest to assess their potential as predictive markers as well. Overall, this study was the first to look at in vivo lncRNA expression in the context of HIV-1 ATI.

In summary, our study explored the expression of lncRNAs HEAL, MALAT1, NEAT, GAS5 and NRON in an ATI context. While the expression of four of the five lncRNAs remains largely stable, HEAL is significantly upregulated at the time of viral rebound. HEAL expression correlation with both HIV-1 dynamics and ISGs during ATI, and the significant effect that knockdown has on HIV-1 infectivity, suggest that HEAL upregulation at viral rebound is directly driven in vivo by viral cues.

## 4. Materials and Methods

### 4.1. Patient Cohort, Sampling, and Clinical Trial

The HIV-STAR ATI clinical trial (clinicaltrials.gov: NCT02641756) was conducted between 1/1/2016 and 1/12/2017, with the aim of characterizing the nature and spatial distribution of the replication competent HIV-1 reservoir by comparing viral sequences from different compartments throughout the body before and after treatment interruption [19]. The Ethics Committee of the University Hospital of Ghent approved the recruitment strategy and experimental design prior to initiation (Belgian registration numbers B670201525474 and B670201628230). All participants provided written informed consent. In total, 12 people (ages between 18 and 65 years) were recruited who met the inclusion criteria: a subtype B HIV-1 infection that was sufficiently suppressed (<20 copies/mL for at least 2 years) at the time of inclusion and normal CD4 counts (CD4 count >500 CD4/µL or >30%) due to long-term ART suppression (2–11 years). All participants converted to the required combinational antiretroviral therapy (cART), comprising 2 NRTIs and an integrase inhibitor, at least 3 months prior to the first sampling. For this substudy, 10 out of 12 patients were retained, as, for two patients, no sufficient amount of biological material was available.

The clinical trial included an ART treatment interruption (ATI), which was considered day 0. Peripheral blood mononuclear cells (PBMCs) were collected prior to ATI at T1 (at undetectable VL), at T2 (7–15 days post ATI, undetectable VL), T3 (viral rebound) and T4 (3 months after cART reinitiation) (Figure 1). Patients were monitored twice a week following ATI to assess VL and potential rebound. Upon rebound, cART was immediately restarted. Viral rebound was defined as a VL > 1000 copies/mL or two consecutive VL > 200 copies/mL measurements.

As control groups, RNA extracted from ART-naïve long-term non-progressors (LTNP, *n* = 17) and uninfected individuals (*n* = 14), originating from a previously published cohort, were included. LNTPs maintained HIV-1 viremia < 1000 copies/mL and CD4+ T cells > 500 cells/mm^3^ over 7 years post-infection. Detailed clinical data and inclusion criteria from this cohort were described previously [16,17]. The Ethical Committees of Ghent University Hospital and of the Royal Free Hospital approved this study (reference numbers: B670201317826 and 13/LO/0729).

### 4.2. Measurement of lncRNA Expression Levels by Quantitative Real-Time PCR

A total of 10^7^ PBMCs were collected per timepoint, and RNA was extracted according to the manufacturer’s instructions (RNA innuPREP Mini Kit, Analytik, Jenna, Germany). The concentration was measured using the Qubit Fluorometer (Invitrogen, Carlsbad, CA, USA) and 1 µg of total RNA was reverse transcribed using the qScript cDNA Supermix (Quantabio, Beverly, MA, USA), following the manufacturer’s protocol. For the measurement of five lncRNA expression levels, 20 ng cDNA was used as input in a 10 µL quantitative real-time PCR (LightCycler 480 II, Roche Applied Science, Indianapolis, IN, USA), while 5 ng input was used to assess eight more abundantly expressed reference genes. The SYBR Green kit (LightCycler480 SYBR Green I Master, Roche Applied Science, Indianapolis, IN, USA) was used according to the manufacturer’s protocol, with 5 µL SYBR Green Master Mix and 250 nM primer concentrations. Each reaction was performed in duplicate. The cycling conditions of the LightCycler 480 (Roche Applied Science, Indianapolis, IN, USA) were 95 °C for 5 min, 45 amplification cycles of 95 °C for 10 s, 58 °C for 30 s and 72 °C for 30 s. Assay specificity was visually assessed in the LightCycler480 software version 1.5, based on the from 60 °C to 95 °C melting curve and all samples passed quality control. Three (GAPDH, ACTB and YWHAZ) out of the eight most stable reference genes were selected with the geNorm method of Vandesompele et al. [28], and the normalization factors were determined by the geometric mean of their relative quantities using the qbasePLUS software 3.1 (Biogazelle, Zwijnaarde, Belgium). All primers used for qPCR can be found in Appendix A. The primer efficiency was assessed on a two-fold cDNA standard curve (50–1.56 ng) and primers with efficiencies between 85 and 105% were retained. The normalized relative quantities (NRQs) of each gene were calculated using the (primer efficiency)^ΔCt^ method. For the control groups, due to the limited amount of available RNA, all but one lncRNA could be checked in RT-qPCR (so NRON expression could not be assessed in this cohort).

### 4.3. HIV-1 Cell-Associated RNA and DNA Measurements

PBMC RNA was also used to measure TAR-, long LTR-, polyA-, Pol- and Tat-Rev-cell-associated HIV-1 RNA transcripts, as described by Yukl et al. [29]. RNA mass recovery was used to maximize the input for reverse transcription reactions and a median of 0.385 and 3.286 μg for (i) TAR and (ii) long LTR, Pol, polyA and Tat-Rev RT was used, respectively. HIV-1 total DNA was extracted from 107 PBMCs using the Dneasy Blood & Tissue Kit (QIAGEN, Germany), and quantified as described by Rutsaert et al. [30] using restriction digestion and droplet digital PCR (ddPCR) including reference gene RPP30 normalization. For integrated HIV-1 DNA, the Alu-HIV-1 nested-PCR method was performed as previously described by Van Hecke et al. [16]. This assay targets the human Alu fragment and the HIV-1 gag region in the first PCR reaction, while only targeting HIV-1 in the second reaction. The unintegrated HIV-1 background was adjusted by assessing it using only the HIV-1 gag primer in the first reaction of the nested PCR.

### 4.4. In Vitro HEAL Knockdown and HIV-1 Infection Experiment

SupT1 cells (NIH HIV reagent program) were cultured in RPMI medium (Gibco), supplemented with 10% FCS (HyClone) and Pen/Strep. They were treated for 48 h with 5 µM of either a scrambled antisense oligo (ASO) (GTGCTGTTCCCGGGGA) or an ASO that targets HEAL (+G+A+GGCTAGCAGGC+T+G+G) with three locked nucleic acids at either side of the ASOs. After 48 h, keeping the ASO concentration at 5 µM, the cells were infected with NL4.3 GFP-tagged HIV-1 lab strain virus by spinoculation for 90 min at 2300 rpm and 32 °C. At 24 h and 48 h post infection, HIV-1 infection was assessed by flow cytometry readout of GFP. The normalized relative quantities of HEAL were determined with quantitative PCR, as described before. All experiments were performed in triplicate.

### 4.5. Statistical Analysis

Normal distribution of the in vivo lncRNA expression data was assessed using a D’Agostino–Pearson test. In case of a normal distribution, a one-way ANOVA was used to assess differences between timepoints (T1–T4), with a post hoc Tukey’s Honest Significant Difference test to specify which timepoints showed differential expression of the lncRNA. In case of non-normal distribution, a Friedman test with a Dunn’s post hoc test was used to detect expression differences among the timepoints. Spearman correlation analysis was used to compute significant correlations between in vivo lncRNA expression and host or virological parameters at single timepoints. A repeated measurement correlation analysis to validate these correlations was performed in R (64 Bit, version 4.0.3) with the rmcorr package. Differences in HEAL expression between ASO treated SupT1 conditions were assessed with an unpaired Student *t*-test.

All statistical analyses and graphical representation were performed in GraphPad Prism9 (9.4.1) and R (64 Bit, version 4.0.3) using the following packages: corrplot, tidyverse, ggpubr, rstatix, rmcorr and ggplot2 [31,32,33].

## Figures and Tables

**Figure 1 ijms-24-01031-f001:**
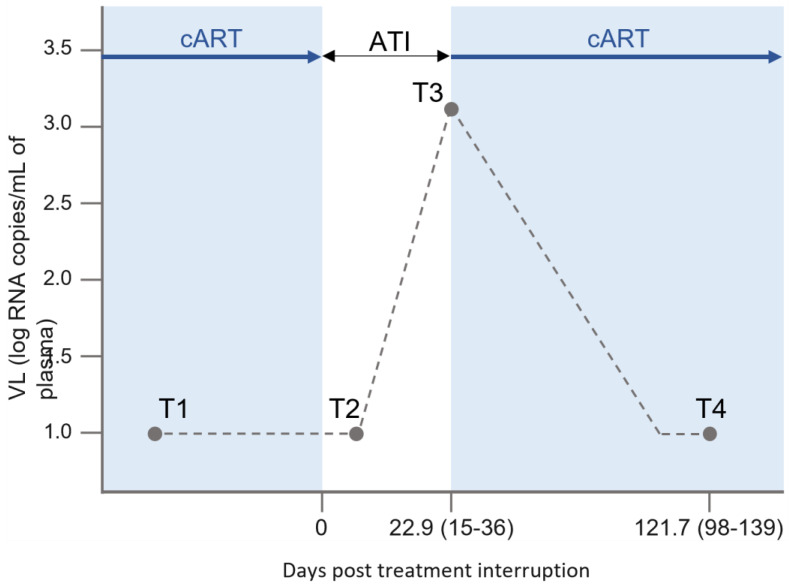
Overview of PBMC sampling timepoints. The clinical trial included an ART treatment interruption (ATI), which was considered day 0. PBMCs were collected prior to ATI at T1 (at undetectable VL), at T2 (7–15 days post-ATI, undetectable VL), T3 (viral rebound) and T4 (3 months after cART reinitiation). Median days for T3 and T4 are depicted, as well as the range.

**Figure 2 ijms-24-01031-f002:**
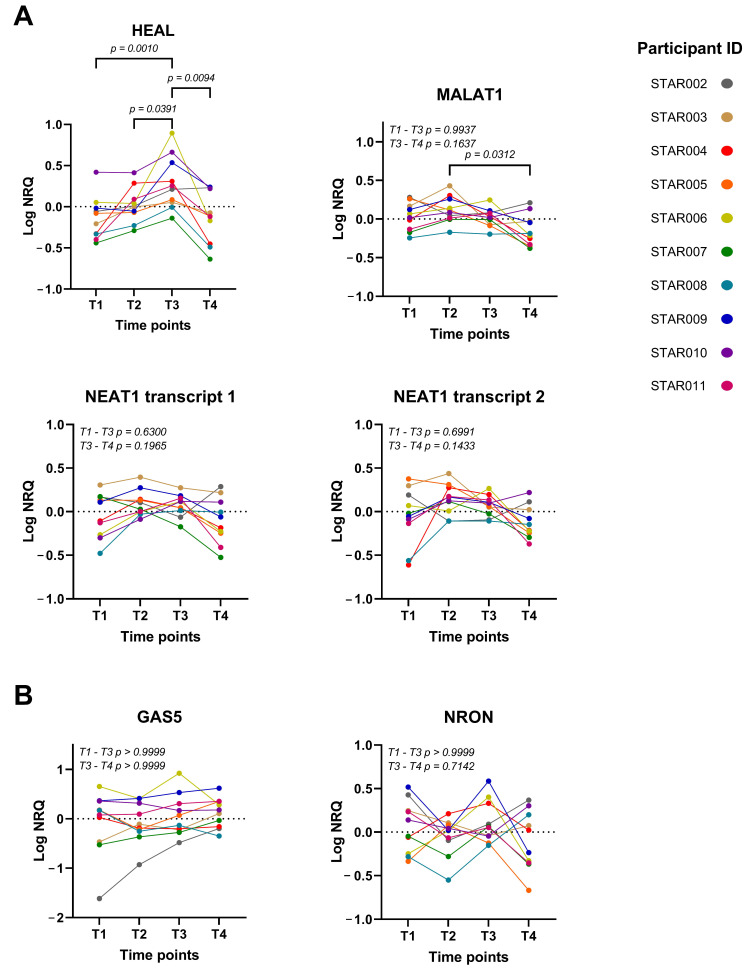
Expression of lncRNAs during ATI in 10 patients on long-term ART. Expression levels of the lncRNAs were quantified with RT-qPCR and normalized using 3 reference genes. Normalized relative quantities are shown at four timepoints (T1-T4) for lncRNAs that are (**A**) up- or (**B**) downregulated upon HIV-1 infection according to the literature. Normal distribution of the data was assessed using a D’Agostino–Pearson test. For normal distributed data, a one-way ANOVA with a post hoc Tukey’s Honest Significant Difference test was performed. For non-normal distributed data, a Friedman test with a Dunn’s post hoc test was performed. Significant *p*-values are indicated with brakes. NRQ, normalized relative quantities.

**Figure 3 ijms-24-01031-f003:**
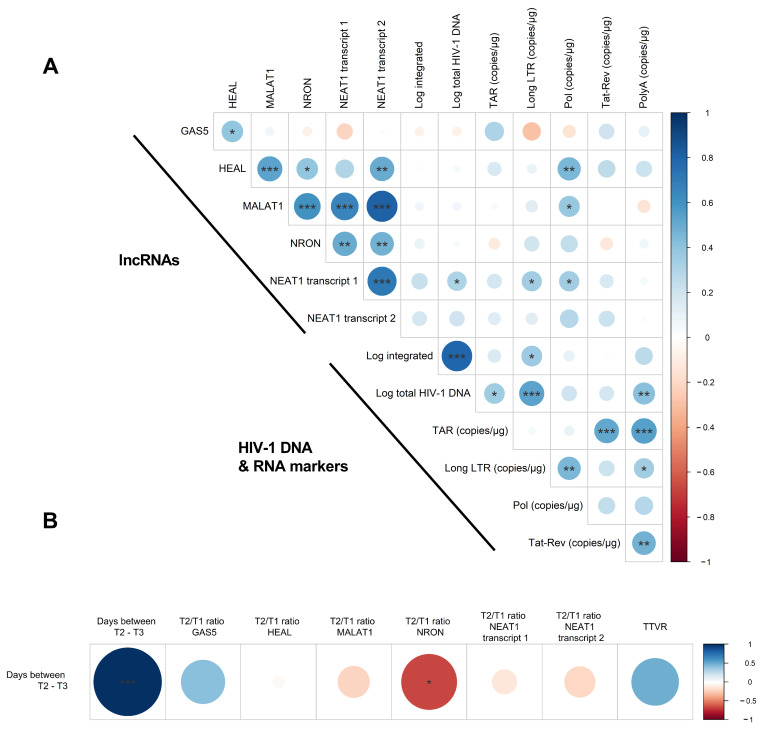
Correlation between lncRNA expression and HIV-1 markers. (**A**) Correlation plot indicating the correlation between lncRNA expression and HIV-1 viral DNA and RNA markers in participants undergoing ATI. Spearman correlations were calculated at all timepoints and included all participants. (**B**) Spearman correlation between lncRNA fold change expression (between T2 and T1) and the days from T2 to viral rebound. Positive and negative correlations are depicted in blue and red, respectively. Significant correlations are indicated with asterisks (*p* < 0.05 *, *p* < 0.01 ** and *p* < 0.001 ***).

**Figure 4 ijms-24-01031-f004:**
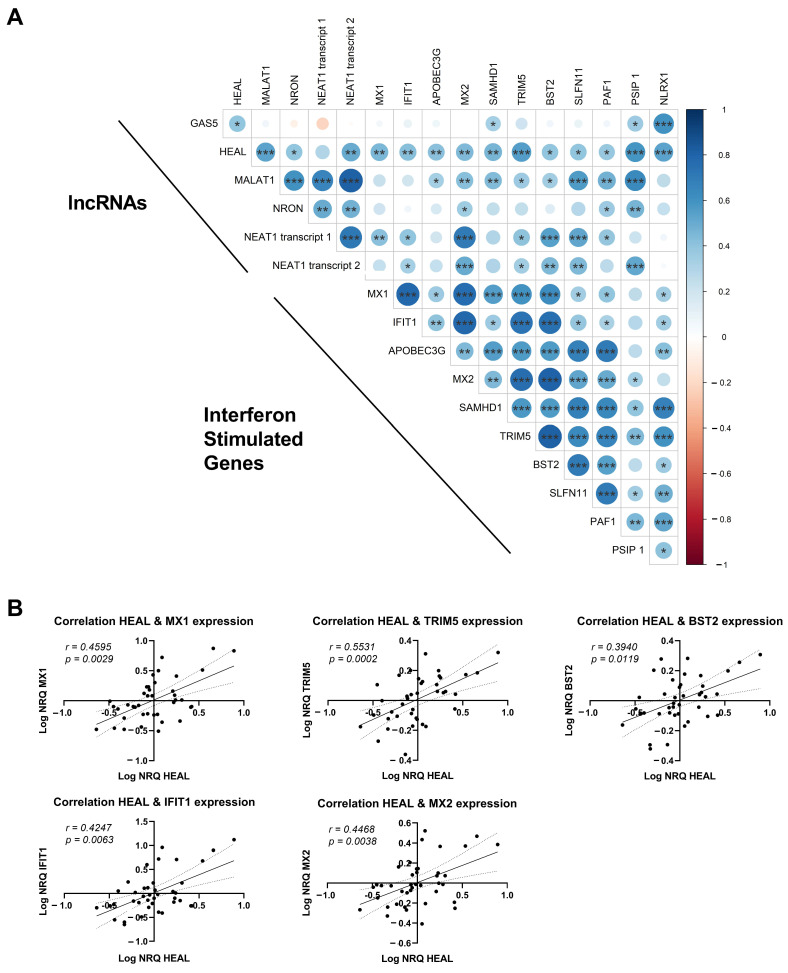
Correlation of lncRNA expression and the expression of interferon stimulated genes. (**A**) Correlation plot indicates the correlation between lncRNA expression and the expression of ISGs in participants undergoing ATI. Spearman correlations were calculated over all timepoints, including all participants. Positive and negative correlations are depicted in blue and red, respectively. Significant correlations are indicated with asterisks (*p* < 0.05 *, *p* < 0.01 ** and *p* < 0.001 ***). (**B**) Spearman correlation plot of the expression of HEAL and restriction factors observed to be upregulated in previous work [18]. In all plots the Spearman r is reported, as well as the *p*-value.

**Figure 5 ijms-24-01031-f005:**
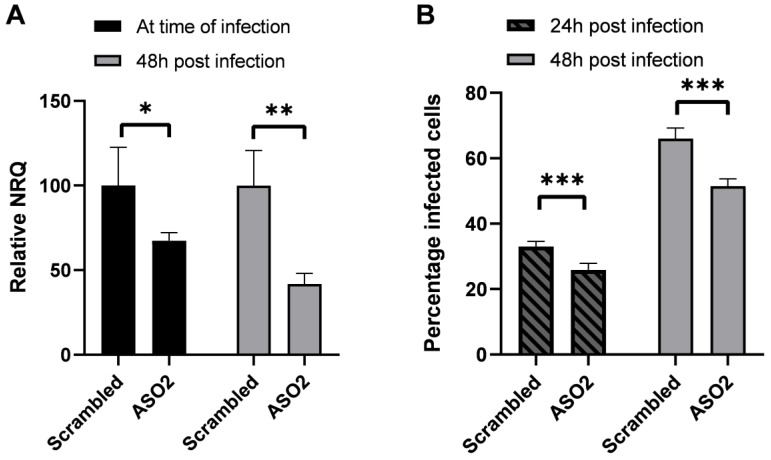
In vitro assessment of the impact of HEAL expression on HIV-1 infection in SupT1 cells. SupT1 cells were incubated with 5 μM scrambled or HEAL targeting ASOs 48 h prior to infection. Cells were infected with NL4.3-GFP virus and monitored for 48 h. (**A**) Relative NRQs were generated for both pre- and post-infection samples. (**B**) Percentage of infected cells was determined using flow cytometry, quantifying the percentage of GFP positive cells. Student *t*-tests were performed to determine significance between two groups (*p* < 0.05 *, *p* < 0.01 ** and *p* < 0.001 ***). All experiments were carried out in triplicate.

**Table 1 ijms-24-01031-t001:** Clinical and virological characteristics of participants in the HIV-STAR cohort; n = 10. Values shown are medians and the interquartile range (IQR) between brackets. VL: viral load; TTVR: Time to viral rebound.

Clinical Characteristics Median (IQR)	
Age (years)	39.5 (37.25–40.75)
Sex	All male
Subtype	All subtype B
Time on cART (years)	3.5 (3.0–7.5)
CD4 nadir (cells/mm^3^)	389 (328–417)
CD4 count at T1 (cells/mm^3^)	746 (618–1000)
CD4/CD8 at T1	1.08 (0.95–1.24)
**Virological characteristics** **Median (IQR)**	
VL zenith (log copies/mL)	4.81 (4.67–5.06)
VL at rebound (log copies/mL)	3.30 (3.26–3.64)
TTVR (>1000 copies/mL) (days)	21 (15–36)

## Data Availability

The data presented in this study are available on request from the corresponding author. The data are not publicly available due to privacy of the participants.

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
