# Peer review of "Evaluating lncRNA Expression Patterns during HIV-1 Treatment Interruption"

_ijms, 2023, doi:10.3390/ijms24021031_

Round 1
Reviewer 1 Report
The paper evaluated long non-coding RNA (lncRNA) expression pattern with 10 patients' PBMCs of a clinical antiretroviral treatment interruption (ATI) trial to explore whether lncRNAs could be potential drug targets and predictive markers.
Major comments:
The authors identified the correlation between HEAL and ISGs, and further showed that knocking down HEAL inhibited HIV infection. However, in Figure 5, A showed the knock down level at the infection point and 48 h post-infection and B showed the inhibition level on the HIV infection.
Please could the authors provide the knocking down level at 24 h post-infection ?
Minior:
Please check the title sentence: "2.3. Correlation between lncRNA expression, HIV-1 markers and TTVR and is limited "
Under Section 2.4 "...expression pattern of IGSs" is that ISGs?
Under Section 2.5 "(figure 5A), (figure 5B)"should be Figure 5A and Figure 5B.
Under the discussion section on page 9 , the fifth paragraph please check the sentence "it is hard to pinpoint wat the dominant driver of the expression of a lncRNA like HEAL is..."
Reviewer 2 Report
This case study reported a clinical trial to evaluate lncRNA expression patterns during HIV-1 treatment interruption. This study was well-designed and well-written. It is interesting and important to the field of HIV treatment and HIV cure and can be helpful for future work. Overall I think this paper had high quality. I had no major comments, but I still have some minor comments:
Minor Issues
1. Language can be improved, especially when describing HIV-related terminologies to avoid stigmatised language, such as HIV-1 infected individuals. Please refer to UNAIDS language guidelines for people living with HIV-1 and have a thorough language check.
2. In the Results section, the authors mentioned all participants were adherent to cART therapy, would it be possible for the authors to reveal which regimen combinations the participants were uptaking?
3. The authors report the median and range in the main text and Table 1, would be helpful to (also) report the IQR of the medians.
4. I would personally move the statistical details of Figure 2 and Figure 5 to the Methods section. Of course, I have no objection if the editor deems it fits here better.
5. It is bearly readable for the details of Figures 3 and 4, please provide a higher-quality figure with larger fonts.
6. In section 2.4, the authors have stated their hypothesis, however, I think this fits better in the introduction.
7. In the discussion section, the authors referred to the evidence from Chao et al., however, no reference was cited.
8. In the limitation section, the authors mentioned that the study has limited generalizability due to the small sample size, I fully agree with the authors. Can the authors also provide a powder analysis for a sample size of 10? Also, all of the participants the authors recruited were male, can the authors also elaborate on the generalizability of their findings to females?
9. In statistic analysis, when referring to the R packages, please also cite the packages to give credit to other authors. Also, in the earlier comment, can the authors conduct a power analysis?
Reviewer 3 Report
The manuscript by Schynkel and colleagues analyzes the expression of five known long noncoding RNAs (lncRNAs, >200 nucleotides) (HEAL, MALATI, NEAT1, GAS5, and NRON) during antiretroviral treatment interruptions (ATI). In this, the levels of these lncRNAs were quantified at four different time points during the ATI. The significant finding of this study is that lncRNA for HEAL was significantly upregulated compared with the other four lncRNAs during the ATI. The manuscript is generally well-written and is suggestive that HEAL expression correlates with a rebound in HIV-1 replication. However, it is unlikely that this could be used as a marker for viral rebound as well-established viral load assays are more specific.
Major comments:
1. The design of these experiments likely has a fundamental flaw. In this, the PBMC of the participants is used to extract RNA samples for analysis. Isolated PBMCs are a mixture of several cell types. These include lymphocytes (CD4+ and CD8+ T cells, B cells, and NK cells), monocytes, and dendritic cells. Thus, it raises questions of: a) What is the expression of these lncRNAs in these cell types and in the presence of ART? ; and b) Could cells other than CD4+ T cells and monocytes be responsible for the increase in HEAL lncRNA?
2. The experimental design lacks a control population. For example, analysis of lncRNA levels from the PBMC of long-term non-progressors (not on ART/ CD4 T cells above 500 cells/µl) would be ideal for this study.
3. Figure 1. The x-axis is not labeled. I assume it is in days. Please fix. The authors state that the T4 time point was at 3 months after reinitiation of ART. Shouldn’t this correspond to 140 days and not 150 days? Also, was the T3 time point at 50 days (i.e., just before the reinitiation of ART?
4. Figure 2. The authors should explain why the lncRNA levels of HEAL at T4 were less than at T1/T2
5. Figure 3A. This figure correlates the five lncRNA levels against various virus parameters. Here the authors state that correlations were made over the four time periods. It is unclear why the data is depicted this way since the lncRNA levels at T1, T2, and T4 are essentially baseline. Why aren’t the correlations made with the T3 data?
6. Figure 3B. This figure correlates the lncRNA ratios from T1 to T2 and supposedly from T2 to viral rebound (I assume T3). However, the correlations between T2 and viral rebound are not shown. Why not? Please fix.
7. Figure 4. In this figure, the authors present a “Correlation of lncRNA expression and the expression of HIV-1 specific restriction and dependency factors.” However, the authors are incorrect in stating that these are HIV-1-specific restriction factors. Many of these factors affect the replication of non-HIV-1 viruses. This title should be corrected. In the legend to Figure 4A, the authors state, “Correlation plot indicates the correlation between lncRNA expression and HIV-1 viral DNA and RNA markers in participants undergoing ATI.” However, there is no presentation of HIV-1 viral DNA and RNA markers. Please fix. Finally, the authors show a correlation of lncRNA expression and different restriction/dependency factors. As antibodies exist to these factors, the authors should not only analyze RNA levels but also protein levels as they are most important.
8. In Figure 5, the authors analyze whether scrambled (control) or anti-sense oligonucleotides (ASO2) would reduce the infectivity of the NL4.3-GFP virus. The authors show modest levels of HEAL lncRNA reduction (~35% of the control at 48 hr) and virus infectivity at 24- and 48-hours post-infection. If the NL4.3-GFP is infectious, the number of cells infected at 48 h hours may reflect the virus spread in the culture. This should be stated.
Minor comments:
1. Page 1, first line: “caused” should be causes
2. Page 1, line 3: please delete “hidden”
3. Page 1, line 4: please delete cause and replace it with “result in”
4. Page 1, second paragraph, line 4: delete “To date”
5. Page 1, last line: change “ [2] and increased levels of transmission [3] to “[2] and an increased rate of transmission [3]
6. Page 2, second paragraph: change HIV to HIV-1
7. Page 2, paragraphs 1-3 and throughout the manuscript: please italicize in vivo
8. Page 3, last line: change timepoints to time-points
9. Page 6, section 2.3: spearman should be capitalized
10. Page 8, last line of the figure legend: Replace “Everything” with “All experiments”
11. Page 9, third paragraph: In the sentence with, “in in vitro cell line experiments,” please delete in vitro as it is redundant.
12. Page 11, section 4.4:In the sentence that begins with 24h, please add “At” before 24h
Round 2
Reviewer 1 Report
Accepted, I have no more comments.
Author Response
We would like to thank the reviewer for his/her work on this manuscript.
Reviewer 3 Report
In the revised manuscript, the authors have addressed most of the previous concerns.
1) However, I believe that the authors should include the data presented in their response. It would provide data that LTNP patients don't have increases in these lncRNAs.
2) I still disagree that comparisons of all time points (T1-T4) in Figure 3 are unnecessary as T1, T2, and T4 are likely baseline.
Author Response
In the revised manuscript, the authors have addressed most of the previous concerns.
1) However, I believe that the authors should include the data presented in their response. It would provide data that LTNP patients don't have increases in these lncRNAs.
We have now included this data as supplemental figure and described this in the manuscript,
2) I still disagree that comparisons of all time points (T1-T4) in Figure 3 are unnecessary as T1, T2, and T4 are likely baseline.
The profiles indeed reveal a peak at T3, but we did not know how this profile would look like in advance, as the other time points reflect different events of the ATI study. Therefore, we used all time points and also included a repeated measurement correlation analysis we added in supplemental.